# Beyond IgE: Alternative Mast Cell Activation Across Different Disease States

**DOI:** 10.3390/ijms21041498

**Published:** 2020-02-22

**Authors:** David O. Lyons, Nicholas A. Pullen

**Affiliations:** School of Biological Sciences, University of Northern Colorado, Greeley, CO 80639, USA; david.lyons@unco.edu

**Keywords:** mast cell, innate immunity, NLRP3, MRGPRX2, inflammatory bowel disease, cancer, food allergy, trained immunity, TGF-β1, IL-10

## Abstract

Mast cells are often regarded through the lens of IgE-dependent reactions as a cell specialized only for anti-parasitic and type I hypersensitive responses. However, recently many researchers have begun to appreciate the expansive repertoire of stimuli that mast cells can respond to. After the characterization of the interleukin (IL)-33/suppression of tumorigenicity 2 (ST2) axis of mast cell activation—a pathway that is independent of the adaptive immune system—researchers are revisiting other stimuli to induce mast cell activation and/or subsequent degranulation independent of IgE. This discovery also underscores that mast cells act as important mediators in maintaining body wide homeostasis, especially through barrier defense, and can thus be the source of disease as well. Particularly in the gut, inflammatory bowel diseases (Crohn’s disease, ulcerative colitis, etc.) are characterized with enhanced mast cell activity in the context of autoimmune disease. Mast cells show phenotypic differences based on tissue residency, which could manifest as different receptor expression profiles, allowing for unique mast cell responses (both IgE and non-IgE mediated) across varying tissues as well. This variety in receptor expression suggests mast cells respond differently, such as in the gut where immunosuppressive IL-10 stimulates the development of food allergy or in the lungs where transforming growth factor-β1 (TGF-β1) can enhance mast cell IL-6 production. Such differences in receptor expression illustrate the truly diverse effector capabilities of mast cells, and careful consideration must be given toward the phenotype of mast cells observed in vitro. Given mast cells’ ubiquitous tissue presence and their capability to respond to a broad spectrum of non-IgE stimuli, it is expected that mast cells may also contribute to the progression of autoimmune disorders and other disease states such as metastatic cancer through promoting chronic inflammation in the local tissue microenvironment and ultimately polarizing toward a unique T_h_17 immune response. Furthermore, these interconnected, atypical activation pathways may crosstalk with IgE-mediated signaling differently across disorders such as parasitism, food allergies, and autoimmune disorders of the gut. In this review, we summarize recent research into familiar and novel pathways of mast cells activation and draw connections to clinical human disease.

## 1. Introduction

Mast cells (MCs) are innate immune cells of the myeloid lineage that are popularly associated with allergic, asthmatic, and anti-worm responses. In the past, research predominantly focused on the IgE-mediated activation of MCs; this mode of activation is dependent on the adaptive immune system to supply antigen-specific IgE to sensitize MCs. Recently, researchers began to focus on characterization of novel MC activation paradigms that are not only independent of IgE-mediated crosslinking but also express unique cytokine secretion profiles. Perhaps the most heavily discussed pathway is mediated through interleukin (IL)-33/suppression of tumorigenicity 2 (ST2) signaling. IL-33-mediated signaling is capable of inducing cytokine expression by MCs, which can also produce IL-33 during IgE-mediated activation but not IL-33-mediated activation [1]. Signaling through this alarmin also synergizes with IgE-mediated responses by increasing MC abundance and enhancing their activation [2]. Not only can MCs respond differently depending on the stimulus, there are also notable differences across MCs based on their tissue residency. Like macrophages, prenatal MCs come from the yolk sac in the developing embryo and are gradually replaced with definitive MCs as the organism matures [3]. These MCs are also phenotypically distinct from one another. In an adult, MC heterogeneity comes from their tissue residence. Although all MCs are capable of producing common T_h_2 cytokines such as IL-4, 5, and 13, their toll-like receptor (TLR) expression and ability to produce renin give those tissue-specific MCs the capability to modulate inflammatory responses and remodel the surrounding ECM [4]. These unique expression patterns manifest differently, and tissue-specific MCs may promote pathologies in a manner unique to their tissue residence. Lung MCs were found to promote bleomycin-induced pulmonary fibrosis through histamine and renin production which promoted wound repair mechanisms and transforming growth factor-β1 (TGF-β1) secretion [5]. Specifically, in the intestines, these mucosal MCs (MMCs) express cysteinyl leukotrienes compared to connective tissue MCs (CTMCs). In addition to this, the expression of P2X7 is present in both intestinal and lung MCs. Both subtypes of MCs also have high TLR expression, further suggesting the MCs in these tissues are predisposed for inflammatory responses [4]. Because of the diversity in MC receptor expression across different tissue types, understanding the microenvironment in which pathology is occurring will lend itself toward developing specific and targeted therapies. Furthermore, the different receptor expression patterns observed across tissue-resident MCs suggest that MCs are specialized for their tissue niche. Across multiple diseases, the phenotypic and morphological changes to the tissue microenvironment may include tissue-resident MCs and thus MCs could be a source of pathology as a result of disrupted homeostatic activity. MCs are experts at initiating and driving inflammation, and their dysregulated activity exacerbates inflammatory conditions in the tissue. Here, we review emerging alternative paradigms for MC activation and discuss their relevance to major gut-related disease states. The possible issue of trained immunity and the paradoxical roles of classical immunosuppressive cytokines are specifically reviewed to stimulate further consideration of these topics specific to MCs.

## 2. Alternative Activation Paradigms

Across all the following inflammatory disorders, MCs are major promoters of pathology and the extent of their activation in these disorders is directly dependent on both the MC’s tissue origin as well as the initiating stimulus. Interestingly, MC activation in the context of these diseases is not solely FcεRI mediated; MCs play a pathological role in an antigen-independent, adaptive immune-independent manner. MCs deserve careful consideration in these gut inflammatory disorders as they are major gut homeostatic mediators. Through IgE signaling, MCs are potent sentinels poised to mitigate helminth threats and unfortunately also drive some allergies. However, we will discuss how they are directly involved in coordinating immune responses and inflammation through their ability to detect and respond to other, non-allergenic stimuli. We open this discussion with a review of such non IgE-mediated signals.

For example, MCs are not only capable of secreting histamine, but they also possess cell-surface histamine receptors, allowing for potential paracrine and autocrine MC signaling in response to histamine release. The expression of the four identified histamine receptors (H1-4R) is specialized for interacting with other cell types and the expression of these receptors on MCs is dependent on the tissue localization of the MC. Human skin MCs were found to express H2R and H4R, and furthermore, these are primarily responsible for mediating the gut immune-microenvironment and signaling with other immune cells, respectively [6,7]. The H1 receptor, which has also been studied for its role in allergic responses, was weakly expressed in normal skin human MCs but was more highly expressed in HMC-1 cells, possibly due to constitutively active c-kit expression in the cell line; this receptor is not hypothesized to be involved in direct signaling in response to histamine [6]. Taken together, the H2 and H4 receptors appear to act as a means of negative feedback on MC degranulation. Expression of these histamine receptors on other cell types suggests MCs can directly signal to other cell types through histamine release. The H3 receptor is a pre-synaptic receptor on neurons which inhibits neurotransmitter and histamine release [8]. These interactions connect the nervous system with the innate immune system; MC-neuronal signaling is largely mediated through substance P (SP) which is released by neurons in response to MC-produced histamine and tryptase; other neuropeptides such as vasoactive intestinal polypeptide can also induce MC degranulation [9,10,11]. In disease states where nervous function may be altered, improper MC-neuronal signaling may induce chronic inflammation without a means for drawing back the inflammatory cell signaling.

Purinergic receptors have also been implicated in mediating MC activation. P2X4 stimulation enhanced prostaglandin E-stimulated degranulation through an alternative mechanism [12]. This enhanced stimulation could be the result of activation of the NLR family pyrin domain containing 3 (NLRP3) inflammasome acting synergistically with typical MC activation paradigms [13].

An emerging unique MC receptor is a GPCR named MAS-related G protein coupled receptor X-2 (MRGPRX2). This membrane and intracellular GPCR is responsible for MC innate immunity and wound healing as well as neurogenic inflammation, pain, and itch. Activation of this receptor results in MC degranulation and is important in mobilizing the adaptive immune system in tissues [14]. Signaling through MRGPRX2 is capable of inducing activation and degranulation of MCs in an IgE-independent manner [15]. MCs are present throughout the body and have also been identified near nerve endings across the skin, gut, and airways [9,16]. Comparable to IgE-mediated T_h_2 signaling, this paradigm can potentiate a MC-dependent positive feedback loop, perpetuating pathologic inflammation in the tissue through exuberant MC activation and cytokine secretion in conjunction with SP release from neurons. SP binds to the neurokinin-1 receptor (NK-1R) and therapeutic inhibition of the receptor reduces symptoms of chemotherapy-induced nausea but strangely does not affect inflammation [17,18]. SP activation on MCs, however, is not mediated through NK-1R but instead through MRGPRX2 in humans or Mrgprb2 in mice [19,20]. Although the mechanism behind this signaling is not yet understood, the ability of MCs to interact with the nervous system suggests their importance in diseases where excessive neuronal activity is present. Such a receptor is not only capable of initiating an inflammatory response without the adaptive immune system, but its ability to mediate MC-neuronal interactions and its relatively exclusive MC expression makes it an attractive target for MC-directed immunotherapeutics. Specific antagonism of the MRGPRX2 receptor is sufficient in inhibiting IgE-independent degranulation and could be used to lessen some drug-induced allergic reactions [21].

The 8-oxoguanine DNA glycosylase 1 (OGG1) is involved in base excision repair (BER) in response to DNA damage, specifically oxidative stress-induced 8-oxoguanine lesions on DNA. These bases excised during OGG1-mediated BER are capable of forming a complex with cytoplasmic OGG1 in the cytosol, changing the conformation of OGG1 and inducing gene expression changes in the MC. These changes promote pro-inflammatory and pro-degranulation gene expression in [22,23]. Interestingly, multiple challenges with 8-oxoguanine resulted in a significant fold-change increase in MC-degranulation-associated genes—these lesions are related to oxidative stress, suggesting inflammasome related signaling may be mediating this effect by priming the MC for enhanced activation through NLRP3 activity; reactive oxygen species (ROS)-induced stress in the gut would likely promote OGG1 activity and could act as a means for promoting a trained immune response.

In addition to these alternative activation paradigms that result in MC cytokine secretion, and sometimes degranulation, we and others hypothesize that MCs possess a form of potentiation unique from adaptive immune system memory. This concept, also known as immune training or trained immunity, has been well described in macrophages but the significance of this training in other myeloid cells, specifically in MCs, has yet to be clearly described. Trained immunity allows for innate immune cells to adapt their response to a broad variety of stimuli, which can protect against future insults; this potentiation manifests as changes in gene expression and epigenetic changes such as histone methylation/acetylation [24]. This response is mediated through detection of pathogen and damage-associated molecular pattern molecules (PAMPs and DAMPs) via pattern recognition receptors (PRRs), namely extracellular TLRs and intracellular NOD-like receptors (NLRs) [13]. The binding of ligands to these PRRs initiates a signaling cascade resulting in inflammasome priming or activation upon re-exposure. There are several inflammasome sensors, each with unique stimuli and diseases associated with their dysfunction. However, most researchers select the NLRP3 inflammasome for study due to its ability to respond to the largest variety of stimuli as well as its two-step activation process [25]. Full activation of the NLRP3 inflammasome induces caspase-1 activity, which cleaves pro-IL-1β and pro-IL-18 to yield biologically active forms. Release of these active cytokines (through secretion or release from damaged cells) results in inflammation and immune cell activity [13]. MCs are not only capable of producing active IL-1β but stimulation with IL-1β is sufficient in inducing histamine release and degranulation from MCs, suggesting MCs can initiate and perpetuate this pyroptotic process in tissues [13,26]. This interaction is also implicated as a potential positive feedback loop for MCs, as histamine can then promote *IL-1* gene expression and synthesis [27]. In the gut, NLRP3 is key in maintaining intestinal homeostasis; NLRP3-deficient mice were more susceptible to ulcerative colitis and displayed reduced IL-1β, IL-10, and TGF-β [28]. The NLRP3 inflammasome is a robust sensor of extracellular threats and is a potent regulator of innate immune responses throughout the body; its role in stimulating trained immunity in myeloid cells highlights the long-term protective effects to a broad variety of pathogens.

In sum, MCs are capable responders to broad immunogenic stimuli. Their response is MC specific and tissue specific; disease states in which these signaling pathways are disrupted demonstrate the unique pathogenic roles MCs can have in the etiology and progression of several inflammatory diseases (see Figure 1). Both the mechanisms of disease and the cellular environment of the affected tissue determine the nature of inflammation. For the remainder of this review, we focus on recent findings in inflammatory bowel diseases, food allergy, cancer, autoimmunity, and autoinflammation, and then we close with special attention to novel effects of IL-10 and TGF-β1.

## 3. Ulcerative Colitis/Crohn’s Disease/Inflammatory Bowel Disease

MCs are pivotal in maintaining gut mucosal homeostasis; inflammatory bowel diseases (IBDs) likely present with defects to MC-related biology. Ulcerative colitis (UC) and Crohn’s disease are the major types of IBDs that arise from chronic inflammation against harmless microbiota; the etiology stems from both genetic and environmental factors [29]. Errors in autophagic responses and polymorphisms, which result in the overproduction of IL-1β and IL-6, have been identified as drivers of chronic inflammation in these diseases; innate immune markers such as *NOD2* and mucin genes are also mutated [30,31,32].

TLR signaling is required in maintaining gut homeostasis and is also important in the clearance of pathogens [33]. Interestingly, MCs in the small intestine weakly express TLR whilst MCs in the colon express high levels of TLR 2 and 4 [4]. The differential expression of TLR could explain why the pathology of ulcerative colitis is limited to colonic tissues compared to more widespread inflammatory bowel disorders such as Crohn’s disease; differential TLR expression could also be explained by the bacterial burden experienced in the small intestines versus the colon [34]. Although the activation of TLRs does not directly induce broad degranulation, ligand binding does lead to cytokine and leukotriene secretion, which promote local inflammation and immune cell recruitment. This change to the surrounding tissue microenvironment likely alters MC activation and response to stimuli such as FcεRI, IL-33, or SP [35]. Across both diseases, innate immunity of the gut is disturbed differently. TLR3 (constitutively expressed in healthy epithelial cells) is down regulated in Crohn’s disease but not UC. However, TLR4 (normally low expression in healthy tissue; receptor for LPS) is highly upregulated across both diseases [36]. In addition to aberrant TLR signaling and expression, NLRP3-related proteins NOD1 and NOD2 were also found to be suppressively mutated in 15–20% of patients with IBDs [37]. The pathogenesis of these diseases arises from a failure of the immune system to quell inflammatory responses leading to excessive, uncontrolled inflammation.

Patients with IBDs exhibit upregulated NK-1R and SP and severity of disease in patients with UC was correlated to levels of SP; rectal SP levels were only increased in patients with UC and not Crohn’s disease and is correlated to a shift in the tissue microenvironment to favor SP production [38,39]. In addition, MCs harboring missense mutations in MRGPRX2 are unable to respond to SP-mediated degranulation—Ca^2+^ mobilization is impacted in this context, which could be the mechanism behind suppression of MCs [40].

Specifically regarding MC activity, patients with UC were found to have greatly enhanced IL-33 expression, and IL-33 producing myofibroblasts were a primary source of IL-33 in these ulcerative lesions and synergized with TGF-β to induce further myofibroblast differentiation; such cells were nearly absent in patients with Crohn’s disease [41]. These IL-33 producing cells in an inflamed gut likely promote MC activation; the protective benefits of this inflammation are likely not present in this disease state and instead exacerbate the inflammatory environment. Beyond the body’s own cells, parasites are capable of mediating the inflammatory responses in the gut. Helminth infestation is inversely correlated with IBDs and clinical treatment using a helminth benefits patients with UC due to the parasite’s capability of inducing a T_h_2 response [42,43]. Patients with UC or Crohn’s disease have higher histamine levels as well as increased H4R expression; MC-produced histamine likely exacerbates the inflammation whilst promoting neutrophilic recruitment and further inflammation [44]. Indeed, these inflammatory bowel diseases are multifaceted and arise as a result of a combination of biological errors and mutations regarding inflammation.

MCs appear to play a greater role in the pathogenesis of UC by promoting a pro-inflammatory microenvironment, mediated by IL-33 and SP-producing cells in the gut. Both diseases present with autoinflammation. However, MCs play a direct role in UC as both signals directly induce MC activation and subsequent TNF-α production in a synergistic manner [45]. Therapies targeting MCs will likely prove more effective in managing the symptoms of UC than Crohn’s disease. However, MCs play a pro-inflammatory role in both diseases.

## 4. Food Allergy

In the context of common food allergies, MCs are essential in the development of allergic responses. This is a case where a typically suppressive cytokine, IL-10, was shown to be necessary for priming mast cells against consumed food antigens (murine oral OVA model); IL-10 was sufficient to enhance IgE-mediated mast cell activation as well [46]. This unexpected response by MCs to IL-10 could be the result of an evolutionary adaptation in response to immunosuppression by parasites—as parasites evolved the ability to suppress anti-worm responses in the gut, perhaps MCs adapted by paradoxically activating in response to select immunosuppressive cytokines. Despite the immunosuppressive nature of parasites, some parasites instead induce an allergic response which can lead to the development of food allergies. Tick bites promote strong allergic responses to the antigen secreted by the tick; the lone star tick induces an allergy to red meat (Alpha-gal syndrome), which behaves like a typical type I hypersensitivity response. IgE production in response to this insult is reliant on TLR stimulation on B-cells through MyD88 [47]. CTMCs in the skin may facilitate the development of such a hypersensitive response through a distinct mechanism rather than typical IgE-mediated food allergy.

IL-33 is also implicated in the development of food allergies. In addition to its direct effects in enhancing IgE-mediated activity, IL-33 promotes oral OVA-mediated anaphylaxis through enhancing MC activity in mice. ST2-deficient mice displayed reduced anaphylaxis. However, the full effect could be reconstituted using WT bone marrow-derived MCs [48]. IL-9 is also implicated in the development of food allergy as well as parasite protection. IL-33 promotes IL-9 and IL-13 production from a unique subpopulation of MMCs; intestinal expression of IL-9 and IL-13 was also increased in atopic patients [49]. These IL-9 producing MMCs were also found to express MC enzymes tryptase and chymase at increased levels compared to healthy patients.

Large extracellular parasites are the eternal rivals of the MC—although the modern world’s developed sanitation and health systems are more capable of extinguishing these infections, gut parasites are still capable of modulating the surrounding immune environment to evade detection. Secretions by *Acanthocheilonema viteae* (ES-62) were capable of suppressing IL-33/ST2-mediated signaling in peritoneal-derived murine MCs; MCs of different tissue residency (specifically bone marrow-derived mucosal MCs) were not as suppressed from the worm byproduct alone, highlighting the tissue specificity of MCs [50]. ES-62 was also shown to be protective against fatal pathology in a chronic OVA/alum asthma model through inhibition of IL-33/ST2-mediated signaling. Parasitic inhibition of MCs varies by parasite and the duration of infection also dictates the immunomodulatory effects of the parasite’s byproducts. Chronic *Litomosoides sigmodontis* infection suppressed intraperitoneal OVA-driven allergic responses and anaphylaxis; interestingly, OVA-IgE levels were unchanged during infection and IL-10 was not found to be involved in the protective effect against anaphylaxis [51]. During infection with *L.sigmodontis*, peritoneal MC counts were found to be decreased, peritoneal MCs harvested were less granular and also exhibited lower levels of histamine [51]. Parasite and worm byproducts elicit changes to the surrounding mucosa by directly altering MC function; such products are parasite specific and could prove therapeutic in dampening excessive MC activity in other disease states.

Though the pathogenesis of this disease is different from typical food allergies, celiac disease (CD) has become more prevalent in modern society. The disease initiates through T-cells in an antigen-specific manner against gliadin within dietary gluten and engages both the innate and adaptive immune systems to induce a proinflammatory response in gut mucosa [52]. MCs are involved in the regulation of the adaptive immune response to dietary gluten and are found in higher counts in celiacs consuming gluten; the higher counts of MCs were found to promote CD severity and progression and MCs were directly activated by digested gliadin fragments [53,54]. The gliadin-mediated activation of MCs in celiacs was also different compared to MCs from a non-celiac patient, highlighting the genetic and environmental basis for CD. Suppressing MC responses and/or replenishment with healthy MCs may slow the progression and severity of CD by restricting adaptive-innate immune signaling and shaping the mucosal environment away from a proinflammatory state.

## 5. Cancer

Among Paul Ehrlich’s original observations on MCs are their association with cancerous tumors; MCs are potent mediators of both pro- and anti-tumor responses in a context-specific manner [55]. As a sentinel cell capable of eliciting inflammatory responses independent of the adaptive immune system, inappropriate MC activation can set the stage for a chronically inflamed environment for cancer proliferation [56]. The production of pro-inflammatory IL-6 and IL-1β can also drive this inflammation in the tumor microenvironment (TME). Recent hypotheses suggest MCs have subsets with unique cytokine profiles similar to macrophage polarization and MCs are capable of switching between the subsets in a context-specific, localized manner, in other words, “tumor educated” MCs [57]. Aggressive triple-negative (ER^−^PR^−^HER2^−^) breast cancers were shown to have higher counts of infiltrating MCs and M2 macrophages mediated by higher expression of annexin A1 [58]. Such immune responses would benefit the progression of cancer through promoting chronic inflammation and wound repair pathways over cytotoxic pathways. MC activity also evolves as tumors progress and expand. In small intestinal cancers, MCs expand in benign polyps in the presence of IL-10, IL-13, and IL-33 as well as ILC2 cells. The presence of these MCs is maintained in an IL-10 dependent manner—overexpression of IL-10 greatly expanded MC populations in these polyps [59]. The presence of IL-10 in conjunction with MC chemokines and alarmins such as IL-33 explains the pro-cancer, pro-inflammatory role MCs can play in certain cancers; in the context of small bowel cancers, MC protease expression resembled MMCs but included CTMC-related mast cell proteases too [59]. When the polyps switched to an invasive phenotype, CTMCs expressing both MMC and CTMC proteases expanded, demonstrating cancer’s ability to alter MC activity toward a pathogenic, pro-cancer function [59]. A mechanism behind this phenotypic switch could be a result of MC interactions with epithelial cells during an inflammatory state, specifically during wound repair. In an azoxymethane-induced colonic tumor model, MCs recruited to epithelial cells during inflammatory wound repair obtained a pro-tumorigenic role and their density in the gut was correlated with cancer grade [60]. Interestingly, the capability of MCs to resolve IL-33-mediated inflammation by damaged epithelial cells was critical in promoting tissue repair following inflammation through protease release. MCs are important in regulating and resolving inflammation within their local tissues; tumors can elicit pro-inflammatory functions in MCs to reprogram them into a pathogenic state. MCs are capable of being activated through IL-33, which illustrates how MCs respond differently to stimulation depending on the surrounding tissue state and current immune status. Additionally, MC-derived IL-6 and TGF-β1 could be considered a pro-tumorigenic threat, as these cytokines can directly contribute to the recruitment of myeloid-derived suppressor cells (MDSCs) and effector T cell polarization away from an anti-tumor T_h_1 toward a T_h_17 phenotype. This likely impacts the efficiency of modern biologic therapies, especially checkpoint inhibiting immunotherapies, where only about 20% of patients respond favorably. It was reported that MDSCs can enhance MC activation, which further suggests a pro-tumor positive feedback loop; however, it remains to be specifically demonstrated whether MCs can enhance MDSC function [61]. Given the presence of MCs across all tissue types, MCs are also implicated as either potential protectors or drivers of cancer. The pro-fibrotic role of MCs in wound repair and inflammation can be dysregulated within a TME and their capability to signal to MDSCs and surrounding fibroblasts can further exacerbate the conditions of the TME. Specifically, in small-bowel cancers, MCs can act as major drivers of inflammation through IL-33/ST2-mediated signaling, promoting chronic inflammation and strongly skewing toward a T_h_17 immune response. Therapies aiming to disrupt MC signaling to the surrounding stroma and immune cells could enhance adaptive immune cytotoxicity and restrict MDSC-mediated activity.

The NLRP3 inflammasome not only contributes to gut-related inflammatory disorders but the resulting chronic inflammation also increases the risk of developing colorectal cancer [62,63]. The NLRP3 inflammasome is largely mediated by downstream apoptosis-associated speck like protein containing a caspase recruitment domain (ASC) and caspase-1 activity. In the context of colorectal tumorigenesis, NLRP3 can play a protective role; ASC and caspase-1 were also found to be protective against tumorigenesis in mice [64]. This connection between NLRP3, bowel diseases, and cancer is multifaceted and the resulting inflammation and cytokine secretion (namely IL-18) can be protective against tumor growth [65]. However, the same cytokines produced following NLRP3 activation are also associated with exuberant inflammation and autoimmune disorders, illustrating both the protective and pathogenic effects of NLRP3. Indeed, the common mutations to NLRP3 and its downstream mediators varies across cancer types and tissue location; the data on these mutations contradicts the protective findings of NLRP3′s IL-18-mediated downstream activity [66]. In addition to acknowledging the tissue specific roles of MC, careful consideration must be made when observing dysregulation of immune mechanisms across different tissue types.

## 6. Autoimmunity/Autoinflammation

Although MC activity alone does not constitute autoimmunity, such activity is sufficient in inducing an autoinflammatory response, in which innate immune cells are activated in response to tissue-specific stimuli [13]. These autoinflammatory diseases can contribute to the pathogenesis of other inflammatory disorders, namely through chronic inflammation. Despite this dichotomy, autoimmune disorders and autoinflammation can coalesce as diseases such as psoriasis and Crohn’s disease, characterized by innate immune activation of T-cells and inflammatory cytokine production [67,68]. Given the biology of MC, their capability to induce an autoinflammatory immune environment through cytokine secretion is only bolstered by their nearly ubiquitous tissue presence and their ability to migrate into so-called immune privileged sites, such as central nervous system parenchyma; and as sentinel cells MCs are one of the first innate immune cells to be activated during an inflammatory response [69]. Consequently, pathogenic activation of MCs is capable of causing harm in privileged tissues.

MCs are hypothesized to mediate autoinflammation through NLRP3 inflammasome sensing of extracellular threats; mutations to the NLRP3 inflammasome and its mediators leads to monogenic diseases such as familial Mediterranean fever (FMF) and cryopyrin-associated periodic syndrome (CAPS) which arise from exuberant caspase-1 activity leading to downstream IL-1β secretion and subsequent inflammation [70].

Due to the ability of MCs to promote and sustain localized inflammation, therapeutic targeting of MCs in autoimmune and autoinflammatory disorders (such as rheumatoid arthritis, UC, or CD) could help to dampen the adaptive autoimmune and innate autoinflammatory responses and promote repair of the damaged tissues. Identifying the soluble factors released in the context of each disease state is critical to understanding how MCs will contribute to the promotion or protection against inflammation.

## 7. IL-10 and TGF-β1

Across these various disease states, MC activity appears to be enhanced, leading to prolonged inflammation and subsequent tissue damage. While most other immune cells are broadly suppressed by IL-10 and/or TGF-β1, MCs react differently. MC responses to typically immunosuppressive cytokines can instead promote MC activation and/or potentiate wound repair pathways and fibrosis (see Figure 2).

TGF-β1 has been characterized as immunosuppressive on MCs through reduced FcεRI expression at the protein level, suggesting subsequently reduced FcεRI-mediated signaling [71]. In terms of phenotypic data, the evidence is conflicting—broad suppression of MC proliferation and activation has been noted [72]. However, some papers present changes in MC inflammatory products based upon the relative differences +/-FcεRI crosslinking; this can lead to the misconception that there is suppression of response when indeed TGF-β1 alone can stimulate MC release of certain factors without concomitant activation through IgE. TGF-β1 may specifically modulate late-phase responses by MCs which could explain the lack of observed effect regarding histamine release or degranulation.

In mice, inhibition of TGF-β1 through a neutralizing antibody caused oral and esophageal inflammation, hallmarked by TGF-β1 producing MCs. Although there was no difference observed in the gut and intestines, the inflammatory response demonstrated by MCs in the mouth and esophagus highlight the role of the MC as a vanguard of innate immunity [73]. Indeed, TGF-β1 is a potent chemoattractant for MCs which implies MCs are equipped to respond; such migration may be critical for wound repair [74]. The TGF-β1-dependent effects observed here also demonstrate how MCs are critical to maintaining homeostasis through tissue-specific interactions. Under pathologic conditions, this signaling axis can skew the local immune response in the tissue. In a murine tumor model, abolishment of TGF-β1 or IL-10 through neutralizing antibodies restored the T_h_1/T_h_2 balance in the tissue, characterized by reduced T_h_2 cytokine secretion [75].

Interestingly, these typically suppressive cytokines can promote a form of protective inflammation mediated by MC activity. In the case of TGF-β1, there is evidence that MCs can activate in response to TGF-β1 in an IgE-independent manner; regulatory T cells (T_reg_) were found to directly enhance MC IL-6 production through surface-bound TGF-β1 and MC cytokine release is enhanced when treated with soluble TGF-β1 [76,77]. Indeed, such atypical activation paradigms potentiate the capability of MCs to impact their surroundings by polarizing toward a T_h_17 environment. In the context of pathology, MCs are capable players in fronting the initial response to cellular injury by shaping the cytokine milieu of the threatened environment. While it is well-appreciated that MCs release IL-6 and T_h_2-polarizing cytokines, the production of (and unique responses to) TGF-β1, IL-17, and IL-22 may reinforce T_h_17-like responses. IL-17^+^ and IL-22^+^ MCs have been reported [78,79] in psoriatic lesions and while the role of an MC-to-T_h_17/22 balance is not clear, the fact that these cytokines are produced by MCs is an important therapeutic consideration, since, for example, MCs can also serve to modulate dendritic cell function [80]. This interaction is also interestingly involved in reducing lung inflammation through suppression of neutrophils and promoting MC IL-6 [81]. The stimulatory response displayed by MCs in these circumstances could be the result of an evolutionary advantage to resist immunosuppression by parasites/worms; MCs may also activate to some degree in the context of wound repair. TGF-β1 stimulation of murine bone marrow-derived MCs is also sufficient in inducing mMCP-1 and mMCP-2 expression facilitated through GATA2 and Smad (2,4, potentially 3) signaling [82]. This response may prime MCs as localized protective effector cells during wound repair or during the resolution of an immune response in tissues. In addition, MC-produced IL-6 is key for clearing bacteria around a wound and allowing for repair [83]. This inflammatory response by MCs may be prompted by TGF-β1 release from the surrounding stroma during an injury, promoting an IgE-independent reaction by MCs without prior adaptive immune priming.

In the gut, IL-10 is a major regulator of homeostasis and is capable of both pro- and anti-inflammatory effects, and like mast cells, its role is context-dependent. IL-10 is critical in providing protective immune cell activation and protective inflammation involved in the development of food allergies and septic defense [46,84]. This protective inflammation is mediated by NLRP3 expression. NLRP3-deficient mice were more susceptible to the development of experimental colitis, reflected by reduced IL-1β, TGF-β, and IL-10 expression [28]. For individuals with exuberant inflammatory diseases such as Crohn’s disease, defects in NLRP3 may lead to pathologic gut inflammation due to the loss of protective tissue-specific inflammation. Conversely, secreted IL-10 in response to NLRP3 activation has also been implicated as a negative regulator of NLRP3 activation; expression of NLRP3 is essential for protective inflammation but unregulated inflammation caused by NLRP3 may also be harmful [85]. In an antigen induced (methylated bovine serum albumin) arthritis model, IL-10 knockout mice displayed more severe symptoms and had higher expression levels of IL-1β, IL-33, and NLRP3 [86]. Non-lethal exposure to endotoxins such as LPS can render immune cells refractory to subsequent exposure and is characterized by reduced macrophage/monocyte cytokine (specifically TNF-α) production [87]. Development of this endotoxin tolerance in MCs has also been shown to be TLR-mediated and associated with a hyporesponsive phenotype [88]. Interestingly, endotoxin tolerance can be alternatively induced alongside TGF-β and IL-10 synthesis in monocytes in response to low levels of toxin; IL-10 suppresses NLRP3 activation during chronic exposure to LPS [85,89]. The downstream effect of NLRP3 activation regarding IL-10 expression is context specific and the timing and duration of NLRP3 activation might also explain the multifarious roles of IL-10 in inflammasome activation and regulation. Specifically, in the small intestines of IL-10-deficient mice, IL-10, TGF-β, and type 3 immune cytokines (IL-17a, IL-22, IL-23) were unaltered. However, IL-33 and IFN-γ concentrations were increased [59]. Progression of polyposis was mediated by MC and T-cell derived IL-10. MMCs expand first in response to small bowel helminth infestation and gradually shifts to CTMC-dominance during resolution of infection [90,91]. In sum, IL-10 is capable of promoting MC activation across multiple tissue types; however, its suppressive capacity must not be overlooked—IL-10 exhibits both pro- [46,59] and anti-inflammatory [84] effects in a tissue-specific and cell-specific manner. Mutations in NLRP3 and IL-10 may help to describe patient susceptibility to inflammatory disease in the gut; prolonged inflammatory diseases likely present with a defect in IL-10 signaling, as chronic stimulation of NLRP3 should engage immune-tolerizing mechanisms through paracrine and autocrine IL-10 secretion. In the context of pathologic MC disorders, dysregulation of both stimulatory and inhibitory paradigms of regulation can indeed promote exuberant MC-mediated inflammation.

## 8. Conclusions/Summary

These alternative activation paradigms highlight the context-specific ability of MCs to mediate the surrounding stroma through cytokine secretion. IL-33 was found to synergize with SP in promoting TNF-α expression; IL-33 was shown to upregulate surface NK-1 expression [45]. MRGPRX2 has been detected in skin MCs and synovial MCs but not lung MCs, suggesting CTMCs may be more susceptible to this signaling and thus be the source of pathogenic inflammation in disease states [92].

Mutations in NLRP3 or dysregulation of signaling may coincide with TGF-β-signaling defects; overexpression of NLRP3 led to increased Smad3 phosphorylation in the kidney, suggesting a pro-fibrotic role [93]. In patients with chronic kidney disease, MCs in the kidney were found to express chymase, tryptase, renin, and TGF-β1 [94]. Expression of chymase is capable of cleaving pro-TGF-β1 into its active form as well as promoting Angiotensin-II activity [95]. The presence of these pro-fibrotic, pro-inflammatory cytokines in the kidney illustrates the context-specific functions of MCs across tissue types. Finally, prolonged activation of NLRP3 and TLR priming can render the MC refractory to future responses [85]. Altogether these observed interactions beg the question of defining an IgE-independent trained immune response in MCs, and whether such training is specific to tissues and/or pathologies.

The timing and context concerning these activation pathways also dictates the suppressive or stimulatory downstream effects. IL-33-mediated signaling on skin MCs transiently potentiates their activation but chronic exposure to the alarmin resulted in suppressed MRGPRX2 receptor expression instead [96]. Pathogenic activation of IL-33/ST2 signaling also occurs in cancer and is associated with an increase in immunosuppressive cell types and increased M2 macrophages. Tumor growth and metastasis was also increased, characterized by the presence of TGF-β^+^ MDSC [97].

Ultimately, MCs are capable promoters of inflammation outside of their typical IgE-mediated role. Therapeutics targeting MC biology should respect the phenotypic differences among MCs originating from different tissues. While MCs may not be the etiologic source of disease, their ability to facilitate inflammation and positively regulate subsequent immune cell interactions/recruitment highlights their pathological capabilities when dysregulated. The fact that MCs express at least one purportedly specific receptor (MRGPRX2) and a relatively specific FcεRI emphasizes a continued interest in these cells as ripe therapeutic targets.

## Figures and Tables

**Figure 1 ijms-21-01498-f001:**
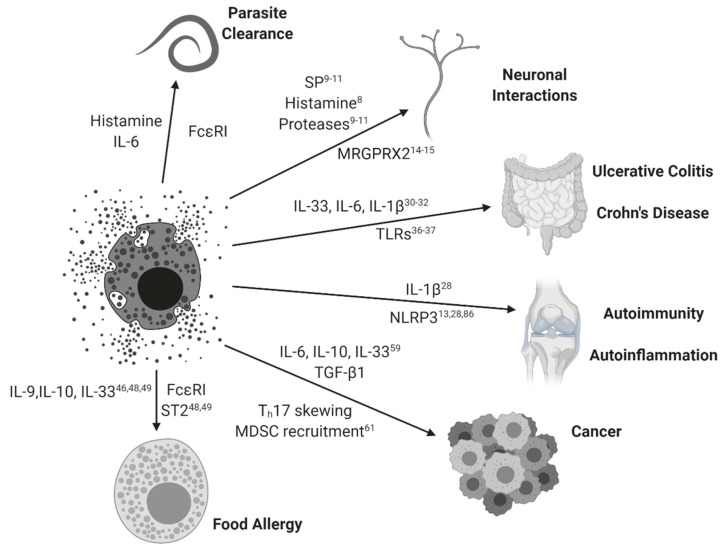
Mast cell activation paradigms. Mast cells and their simplified interactions with clinical diseases are represented with arrowed connections. Relevant MC-mediated soluble factors and associated signaling pathways are represented for each disease state. Subscripts correspond to relevant references from this review.

**Figure 2 ijms-21-01498-f002:**
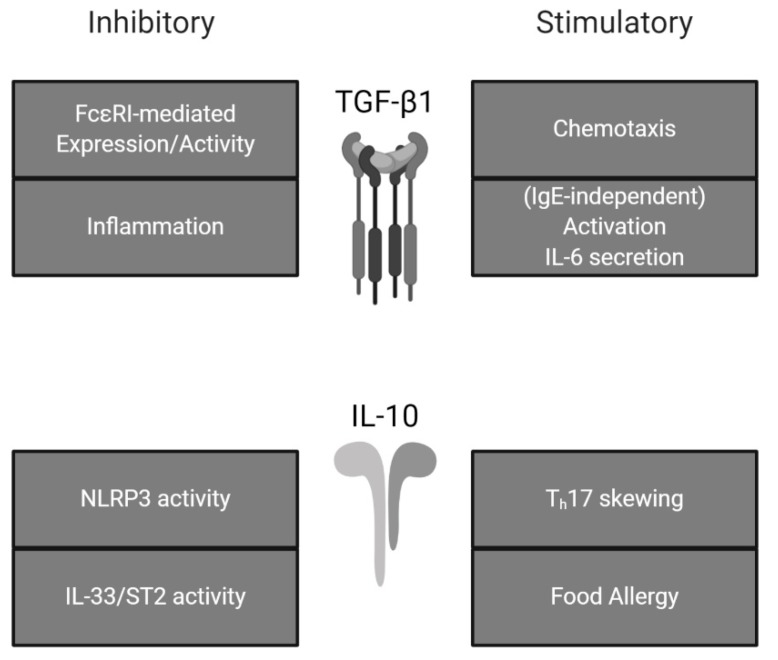
Effects of TGF-β1 and IL-10 on mast cell activity. TGF-β1 and IL-10 exhibit both stimulatory and inhibitory effects on mast cells. Summarized from the present review.

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
