# Peer review of "Beyond IgE: Alternative Mast Cell Activation Across Different Disease States"

_ijms, 2020, doi:10.3390/ijms21041498_

Round 1
Reviewer 1 Report
This is an excellent and timely review on mast cells and alternatively-induced pathways of mast cell activation. The manuscript has been considerably improved since the previous submission, incorporating comments from the Editor and the reviewers.
Minor comment:
While the final section on IL-10 and TGF-b is aimed at discussing the balance of pro and anti-inflammatory effects of these cytokines on mast cells, in the case of IL-10, it comes across as anti-inflammatory heavy (even though the authors discuss this in preceding sections). Lines 435-437 are confusing. Considering that a major goal of the review is to highlight novel pro-inflammatory roles for these cytokines on mast cells, the authors could consider adding a couple sentences in the discussion re-stating the major findings of references 46 and 59 for the reader, so as to provide appropriate context. Basically you could just restate lines 435-437 as: IL-10 is critical for the activation of mucosal mast cells during food allergy and small bowel cancer as described above (46,59) and has been shown to have both pro- and anti-inflammatory effects in patients with sepsis (84).
The TGF-b-related discussion is well-balanced.
Author Response
We thank the reviewer for their time spent addressing this version of our manuscript, and for their positive comments and encouragement. We agree that the summary sentence was confusing and cumbersome to read. We have re-written it per the suggestion.
Reviewer 2 Report
This is an excelent review very well written and comprehensive. The discovery of the recent characterization of the IL-33/ST axis of mast cells activation independent of IgE pathway opens new perspectives to enlarge their role in many physiological and pathological situations. Other interesting aspect is the different response of these cells in several organs and diseases actuating directly, or through IgE mediation response. Both the mechanisms of the disease and the cellular environmental determine the nature of the inflamatory process. So, the final response and the mediators involved in different tissues and pathological situations are different.
The authors describe clearly the involved mediatos and the response in diferente situations such as the parasite clearance, the neural interactions, the role in IBD patients (Ulcerative colitis and Crohn´s disease), the autoimmunity and autoinfammation proceses, and also in different types of cancer and finally in food allergy
All of these disease can be treated and controlled designio new drugs that can modulare the inflammatory response mediatad by the mast cells.
Interestingly mast cells express weakly TLR in the small bowel and by contrary are in high leves in the colon. Across UC and CD the response are also different because the innate immunology of the gut is disturbed in diferente degree and way
The references of this paper are good an actual
In summary this review is rich in containt, very actual and useful for all the readers
Author Response
We thank the reviewer for their thorough reading of our manuscript, and their positive comments and encouragement.
This manuscript is a resubmission of an earlier submission. The following is a list of the peer review reports and author responses from that submission.
Round 1
Reviewer 1 Report
The review article by Lyons and Pullen tries to summarizes recent progresses in the area of IgE-independent MC activation and its roles in diseases, a highly timely topic. Obviously, there are many papers which appear to be not well-connected or difficult to connect to the theme. Their effort is only half successful, as I did not feel that I have understood what is written after my first reading. After reading the manuscript twice, I think this may be due to the following issues:
Some terms are not defined well. For example, in the Introduction, line 51, no well accepted term such as “canonical MC cytokines” exist to my best knowledge. Line 134, OGG1 is introduced without any explanation. “ROS” in line 141, “triple-negative breast cancers” in line 250, “hermetic” in line 288, and “endotoxin tolerance” in line 362 should be spelled out. “innate memory” in line 163 and PAMPs/DAMPs” in line 148 should also be defined. A table will be useful, which describes Disease entity, factors showing MC involvement, and references. A table or figure will be useful that summarizes functions of TGF-b1 and IL-10. The first 2 paragraphs in Conclusions/Summary can be moved to earlier sections. Line 9: Delete “thus responsible for”. Line 86: Add “of” after “production”. Line 91: Should “extracellular” be replaced with “cell-surface”? Lines 135-136: “forming a complex with OGG1” is not clear. Lines 163-164: “the concept of conferring……in an inflammatory setting.” How is this sentence linked to the above descriptions? The description in Section 6 is vague.
Author Response
We thank the reviewer for their time reading our submission and appreciate their substantive feedback. Our updates can be clearly found in track changes (line numbers below are by the track changes count). We agree that the use of some terms and abbreviations required additional explanation, definition, and context. We have made the following changes:
Removed mention of any cytokine set or other activity as being “canonical” including the instance highlighted by the reviewer and two additional uses of the word. (lines 51, 125, 403) New sentences defining and introducing OGG1, along with relevant references. (lines 136-140) ROS, triple-negative breast cancers, and PAMP/DAMP are spelled out. (lines 144-145, 156) Endotoxin tolerance is defined with relevant reference. (lines 425-427) The word hermetic was removed. (line 344) The use of the term “innate memory” has been removed from the document as it is more accurate to describe this type of potentiation as “trained immunity”. (lines 148, 151-152, 172) Figure 1 has been updated to include citations relevant to the interactions depicted. A new figure 2 has been added to section 7 on IL-10 and TGF-b1 We have made all other specific additions and deletions suggested for grammar and clarity (g., extracellular to cell-surface).’ Lines 368-374 have been updated to reflect more specific discussion of autoimmune-to-autoinflammatory involvement of mast cells.Reviewer 2 Report
This is an excellent and much-needed review on the current state of knowledge of non-IgE-mediated events in mast cells, tissue specificity of this reactivity, and their roles in disease. The article is clearly written and provides ample support for its thesis. I don't have any recommendations that would make this piece stronger than it already is.
Author Response
Thank you for your time reviewing our manuscript, and your positive comment.
Reviewer 3 Report
This is a very interesting review that clearly explains and describe the real situation state of the MC in many different situations.
I want to ask some questions to the authors and if it is possible to clear :
1/. I would like to receive more information about the different rol of MCs in UC and Crohn´s disease in the pathogenesis of both diseases and the suggestions about a potential therapeutic effect in the future. In what of the two diseases do you think would be more useful to implant in the clinic?
2/. I would like that you include some comments about the relationship with the Celiac Disease, because it is a common intestinal and autoimmune disease
3/. In which parasitic diseases are mainly involved the MCs and what are the mediators on there
4/.- I would like to have more comments about the list of autoimmune disorders that you think are more implicated and the future indications to use as a therapy
5/ And also to concrete in what tumors do you think are more closed in the pathogenesis and the future indications on therapy
Author Response
We thank the reviewer for their time and thoughtful review of our manuscript. We appreciate the very important suggestions and feel that they have helped us improve the paper substantially. Specifically, we have done the following (which are clearly marked in track changes; line numbers are with track changes on):
Added prose and a reference describing the application of mast cells in ulcerative colitis and/or Crohn’s disease with an additional citation. (lines 231-236) Added additional prose and references detailing Celiac Disease and the involvement of mast cells in the pathogenesis of the disease. (lines 277-288) We have described additional unique parasites and their mechanisms for suppressing mast cell activity and described some potential therapeutic applications that could arise from that parasite’s byproducts. (lines 266-276) Lines 368-374 have been updated to reflect more specific discussion of autoimmune-to-autoinflammatory involvement of mast cells. We have edited the prose in lines 326-334 (cancer section) to be more specific.Reviewer 4 Report
This is a well-written, excellent review emphasizing IgE-independent pathways of mast cell activation and consequently their functional profile across various inflammatory disease states. Overall, the authors do a pretty thorough job describing the various ways in which mast cells can be alternatively activated and contribute to disease pathology.
1) Please make sure that any abbreviations are correctly defined prior to first use.
2) Although the focus is on unique alternatively-induced pathways of mast cell activation, some areas could benefit by further expansion to provide better context. For example in the food allergy section, as the authors talk about IL-33 and mast cells in the preceding section, they could also slightly elaborate on IL-33-mediated pathways of mast cell activation during food allergy and describe the roles of cytokines such as IL-9, etc.
3) Please be consistent with the use of MC vs. MCs. In several places, the authors mean MCs, but use the singular.
4) Please check the manuscript for missing sentences/phrases as in lines 73-74.
Author Response
We appreciate the reviewer’s time spent reading our paper, their thoughtful points, and positive comments. Specifically, we have addressed the comments as follows (which are also clearly marked in track changes; line numbers are with changes on):
We have carefully reviewed the manuscript for definitions and fixed several instances of unexplained acronyms. We have added a new section discussing IL-33 and IL-9. We agree that these are very important additions to this review (lines 252-259). We agree that consistency is necessary with the MC acronym, so we have carefully checked and corrected this where necessary. Mast cells are abbreviated as (MCs) throughout the manuscript. Instances of MC are only used when clearly referring to the singular case. Thank you, we have corrected these errors.